# Plant parentage influences the type of timber use by traditional peoples of the Brazilian Caatinga

**Kamila Marques Pedrosa**[1]\*, **Maiara Bezerra Ramos**[1], **María de los Ángeles La Torre-Cuadros**[2]\*, **Sérgio de Faria Lopes**[1]\*

1 Laboratório de Ecologia Neotropical, Departamento de Biologia, Centro de Ciências Biológicas e da Saúde, Universidade Estadual da Paraíba, Bairro Universitário, Campina Grande, Paraíba, Brasil, 2 Universidad Científica del Sur, Villa EL Salvador, Perú

\* kamila_biopb@hotmail.com (KMP); mlatorrec@cientifica.edu.pe (MALTC); defarialopes@gmail.com (SFL)

**Data Availability Statement:** The data generated and analyzed for the current study has been uploaded to the Dryad Digital Repository https://doi.org/10.5061/dryad.w3r2280w5.

## Abstract

Local populations select different plants to meet their demands, so that morphologically similar species can be more used for a given use. Herein, we seek to understand whether plant species that are phylogenetically closer together are used more similarly than distant species in the phylogeny. Ethnobotanical data were collected in five rural communities in a semi-arid region of Brazil. A total of 120 local experts were selected and interviewed using semi-structured questionnaires. The people's knowledge of plants was organized into usage subcategories. We estimated the redundancy values for the mentioned species, and we compiled data from the literature on the wood density values of the cited species. We constructed our phylogenetic hypothesis of useful plants and used comparative phylogenetic methods to estimate the signal. Our results showed a strong phylogenetic grouping for both tool handle and craft uses. We observed a moderate phylogenetic grouping in which related cited plants exhibit similar redundancy and a weak grouping in which cited plants present similar wood density values. Our results revealed the importance of using phylogeny for useful plants. We conclude the phylogenetic proximity of useful plants and the lower redundancy for some species in our study may suggest greater use pressure, given that few species fulfill the same function.

## Background

Plants and human beings are involved in a dynamic process, as human beings developed ways to use them based on their knowledge, practices and beliefs [1]. These practices are one of the most tangible proofs of the reality of ecosystem services [2], and the provision of wood from plants by local populations of the Brazilian Caatinga is one of the main resources used for producing rural artifacts, biofuel and construction [3, 4]. Thus, investigating the local knowledge of the people about the plants used by them is interesting and relevant, considering that the rural populations who live in the Caatinga use wood as a raw material to meet their daily demands.

**Funding:** MBR and KMP thank the Fundação de Apoio à Pesquisa do Estado da Paraíba (FAPESq) for granting research grants. This study was partly funded by the State University of Paraíba grant 03/2022. SFL thanks CNPq for the productivity grant.

**Competing interests:** The authors have stated that there are no competing interests.

Despite all the diversity of plants that occur in the Caatinga (a Seasonally Dry Tropical Forest) [5], human groups use a specific subset of species to meet their local demands [4]. To the best of our knowledge, less than 70 species of timber plants have been described in the literature as useful by local human populations in the Caatinga [6, 7]. The selection of these useful species is generally influenced by their availability in the environment [8, 9] or by the wood quality of the species [10, 11]. Local populations observe and select species according to their abundance and frequency in the natural environment. For example, it is common for people to randomly use plants to prepare biofuels due to the high demand for biomass or the availability of species in the plant community [8]. While in the second case, people follow criteria to select resources; for example, people look for attributes, wood density, of plant species that give greater performance during their use [11–14].

Plant species tend to embody similar functional attributes [organism characteristics] due to evolutionary heritage [15]. Thus reason, ethnobotanical studies use phylogenetic hypotheses to answer whether useful plant species share similar uses [16–18], suggesting that species which are more closely related to each other would have similar morphological traits [19]. This does not imply that phylogeny influences the traditional use of plants *per se*, but that traits of useful species may be more similar than expected at random, as related plants are similar in both morphophysiological traits and utility [20]. Thus, it is to be expected that the use of plants by humans is influenced by the characteristics of related species, such as the size of the plant [21], but it is still not very well understood what the characteristics of Caatinga plants which support timber uses are, and whether these similarities are related to phylogeny.

Associating characteristics of certain species can serve as a strategy for plant selection by human groups. Among these strategies, utilitarian redundancy is an important phenomenon among groups in the Caatinga [22, 23]. Utilitarian redundancy predicts low pressure on the usage of species based on the idea that the greater the number of species indicated for the same use, the lower the usage pressure on them, except when there are preferred species [22, 24] and phylogenetically related species [18]. A recent study of evolutionarily-related medicinal plants found that the effect of preference on usage pressure depends on species redundancy [18]. This result may concomitantly change the scenario regarding the understanding of species' usage pressure, as related plants may be more used by human groups [18]. However, despite previous studies addressing this issue, it is still necessary to understand the similarity in the functions of certain redundant species that are selected for timber uses.

In this context, our objective is to understand whether related species are known for the same purpose by local populations in a region of the Caatinga. We start from the premise that phylogenetically closer plant species share morphological characteristics and are used more similarly than distant species in the phylogeny. Assuming that the availability of plant species in the natural environment influences the collection of plants for firewood and construction [8], we expect a positive phylogenetic signal in the traits shared between the species used, except for firewood and fencing. Furthermore, given that plant species are evolutionarily related [25], we seek to answer: (i) whether there is a phylogenetic signal for the wood density of the cited species; and (ii) if the phylogenetically related species cited show similarity in redundancy.

## Methods

### Study area description

The present study was conducted in a portion of the Brazilian semi-arid region (Cariri region), in the state of Paraíba, Brazil. The study involved local human populations located in five rural communities distributed at different points (Fig 1). The rural communities involved were

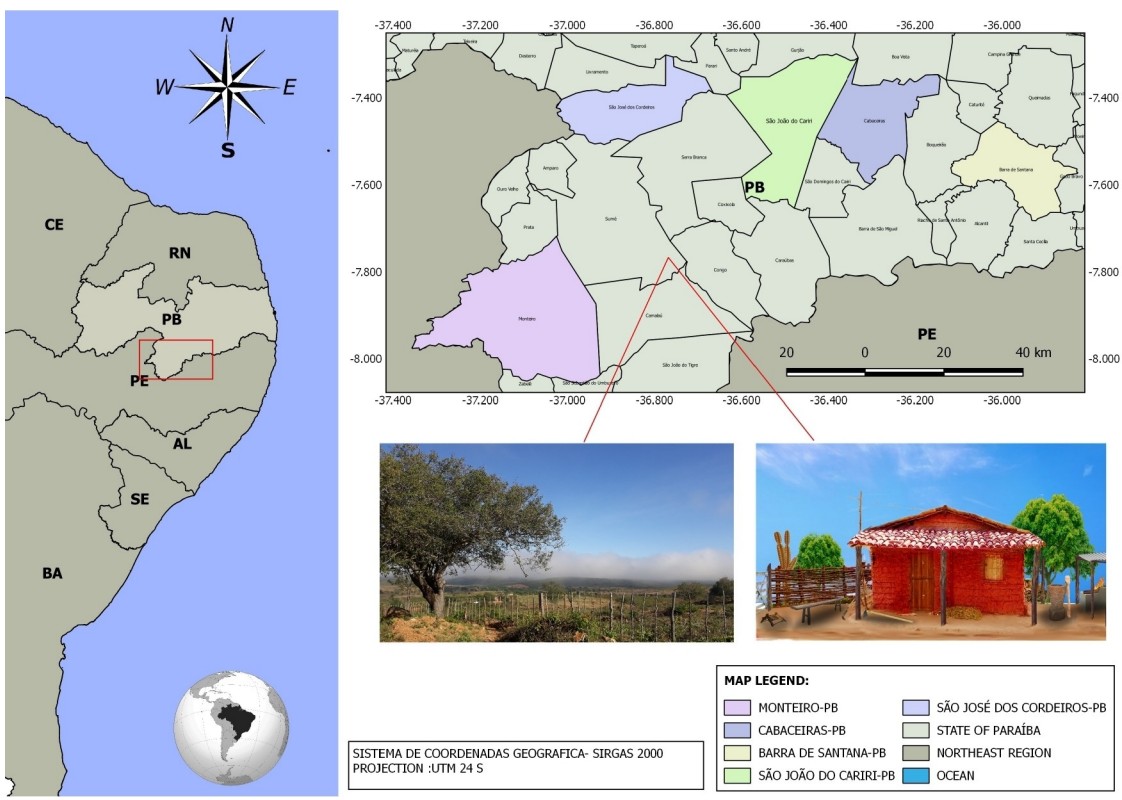

**Fig 1. Location of the studied communities, Paraíba state, in northeast Brazil.**

Riacho Fundo (7°23'47.6"S, 36°27'04.5"W), located in the municipality of São João do Cariri; Caiçara (07°23′8.12″S, 36°23′36.74″W), located in the municipality of Cabaceiras; Alto dos Cordeiros (7°30'53.6"S,°59'46.6"W), located in the municipality of Barra de Santana; Viveiro (7°25'33.5" S, 36°48'43.0"W), located in the municipality of São José dos Cordeiros, and Olho d'água (7°54'56.6"S, 37°17'09.2" W) located in the municipality of Monteiro (Fig 1).

The people who live in rural areas of the Caatinga are small farmers, also known as *catingueiros* [26]. The *catingueiros* are of mixed origin, descendants of older groups of indigenous peoples, *quilombolas* (a quilombola is an Afro-Brazilian resident of quilombo settlements first established by escaped slaves in Brazil) and European settlers, the result of the European colonization process that expanded throughout the 16th century [27]. The *catingueiros* located in the Cariri region of the state of Paraíba, Brazil, are culturally similar [24]. Given the climatic conditions with only two seasons (rainy and dry), it allows most local populations to raise small herds of cattle and goats and carry out agricultural practices such as planting grains, beans, corn, and vegetables in the rainy season.

The Caatinga climate is Bswh (hot to hot-semi-arid, according to the Koppen classification), and stands out for having one of the lowest rainfall regimes in Brazil, ranging from 350 to 600 mm per year [28]. The average annual temperature varies between 25°C and 41°C. The Caatinga is considered a Seasonally Dry Tropical Forest (SDTF) and its vegetation is influenced by environmental filters, climate, soil, temperature, which modulate the assembly of plant communities [29]. At local scales, environmental variables such as topography, terrain slope, soil moisture and anthropogenic disturbances also play key roles in community building in these ecosystems [30]. The Caatinga is represented by endemic flora, endowed with a set of

morphophysiological adaptations to climatic conditions, and a strong phylogenetic niche conservatism [31], resulting in highly heterogeneous vegetation types.

## Ethical and legal aspects

This study was carried out in accordance with the guidelines required by the National Health Council of Brazil through the Research Ethics Committee (resolution no. 466/12/CNS/MS; project approval protocol: 30657119.3.0000.5187). In addition, we obtained the consent of the participants through their signing of an Informed Consent Form (ICF). We requested authorization from the Chico Mendes Institute for Biodiversity Conservation (ICMBio/SISBIO), an agency linked to the Ministry of the Environment (MMA) (registration: 73540–1) to collect the botanical material. The species were identified through specialists according [32]. All species were incorporated into the Manuel de Arruda Câmara Herbarium, State University of Paraíba, Campus I, Campina Grande, Paraíba, Brazil.

## Participant selection

We selected research participants using the snowball technique [33]. The Snowball method is when a participant recommends citing another participant of similar competence, repeating the process until saturation or until the desired sample size is reached. Thus, men and women who practice subsistence activities in agriculture and raising domestic animals were selected. Participants under 18 years of age were not included. Data collection took place between August 2018 and February 2020 (data collection took place twice a week on average). We conducted interviews with 120 local experts (men and women) residing in the five selected rural communities. The local experts represent a sample of the local human population who livein the rural communities of the present study. Thus, the results and conclusions of the study will be from the perspective of the local knowledge of this sample.

## Collection of local ecological knowledge

We used semi-structured forms to collect information about local knowledge containing the following questions: Which plants do you use? How do you use plants? What plants are used for rural construction and home construction? What plants are used as firewood? What plants are used for coal? Which plants are used as technologies (production of artifacts through specific treatments). Usage citations were organized into categories and subcategories according to the ethnobotanical literature [34], namely: fuel (firewood), construction (wire fencing, hedge fencing, roof and door) and technology (tool handles, household items and crafts).

A total of 120 people participated in the study, represented by: Riacho Fundo community = 30 (16 women and 14 men, with an average age of 58 years); Caiçara community = 32 (20 men and 10 women, with an average age of 55 years); Alta dos Cordeiros community = 24 (19 men and five women, with an average age of 45 years); Viveiro community = 23 (15 men and eight women, with a mean age of 41 years); and Olho d'Água community = 11 (seven men and four women, with a mean age of 48years).

Most people are engaged in agriculture. Traditional corn and bean crops generally take place during the rainy season. Small herds (goats, cattle, pigs and sheep) are raised which generates the main local income. The communities are assisted by federal government social programs.

## Estimating the redundancy value of the cited species

The species redundancy proposed by [18] was calculated for each species cited by local experts during the interviews. The index consists of $R = \sum si/n$, in which: $\sum Si$ is the sum of the total

number of plant species that can be used for a given use; $n$ is the total number of species cited by the participants; and $W$ is the total number of uses of a given species. However, the "W" component of the index tends to increase the weight of the versatility of the cited species, which consists of the number of usage categories indicated for a plant [35]. Therefore, we chose not to use this component in our analysis and adapted it by removing it.

## Wood density of the cited species

We collected a morphological measure of wood density (WD) which is related to wood quality for all plant species cited by the local experts in our study. The data referring to the wood density of the species were collected from a literature search. We consulted "Scielo" and "Google Scholar" using the keywords "Scientific name of the species + "wood density" to collect the information (S1 Table). In addition, we extracted data on some species from the Global Wood Density Bank [36]. We assigned the value corresponding to the genus for the species that we did not find the values for wood density. Only the species and genera found in the Caatinga biome were recorded. Moreover, we did not collect data that mentioned common names without the scientific name of the species. We focused on tree plants because they are the most useful in terms of timber use for populations in the Caatinga.

## Phylogenetic tree

We constructed a phylogeny of timber plants using the V. PhyloMaker package in R [37]. This tool provides a phylogenetic hypothesis for vascular plant relationships [37]. The megatree includes all plant families, approximately 10,587 genera, 70,533 species and 479 families of vascular plants existing in the world [37]. The phylogenetic hypothesis we used was constructed using molecular data from GenBank, phylogenetic data from the tree of life, and the fossil record [37]. V. PhyloMaker provides three types of phylogenetic trees; we used S3, which uses the same approach implemented in Phylomatic and BLADJ.

Next, we standardized the nomenclature of plant names according to The Plant List (www.theplantlist.org) and flora Brasil (http://reflora.jbrj.gov.br/reflora/listaBrasil/) through the plantiminier repository (package gustavobio/flora) in order to build our phylogenetic hypothesis. We chose to use both platforms to obtain greater data fidelity. We built a phylogenetic tree considering the species that were cited by the local participants, excluding Cactaceae, Arecaceae and unidentified species (genus or family), and we considered 39 genera and 17 botanical families for the study.

## Data analysis

Prior to the analyses, we coded plant uses as binary traits: 1, when a species is used for a timber use from the seven categories, and 0 when no timber use was assigned to the species (S1 Table).

We used Pagel's lambda (henceforth λ) [38] to test our first hypothesis that the characteristics of species used by local populations are suitable for similar uses, in which we assume that the evolution of traits follows the Brownian motion model [25], where: values of λ close to 0 are found for data without phylogenetic dependence, while values close to 1 indicate strong phylogenetic clustering.

We also used Pegel's lambda for our questions that phylogenetically close species tend to have species redundancy and similar wood density. Furthermore, considering that each wood use has different quality requirements [39], we used Pegel's lambda to verify whether phylogenetically close species tend to have similar wood density for each use subcategory. We also evaluated the correlations between species redundancy and wood density of the cited species

through a phylogenetic autocorrelation using a generalized phylogenetic least squares (PGLS) analysis [40]. All analyzes were conducted in R [41] using the phylolm [42], phytools [43], geiger [44] and caper packages [45].

## Results

We recorded 44 plants belonging to 17 families and 39 genera (Table 1) from a total of 120 people interviewed. Most families had only one or two species registered for some use, with the exception of Euphorbiaceae and Fabaceae, with six and 20 species, respectively. Wood was the main plant part indicated for uses, except for the hedge fencing category, in which the entire tree is used. The highest number of species was indicated for wire fencing (32 spp.), followed by firewood (28 spp.) and tool handles (21 spp.).

The uses with the lowest number of species mentioned were roofs (11 spp.), doors (10 spp.), hedges (eight spp.) and household items (five spp.). Among the species, *Tabebuia aurea* (Silva Manso) Benth. & Hook. f. ex S.Moore had the highest value for wood density, while *Cenostigma pyramidale* (Tul.) Gagnon & G.P. Lewis L. had the lowest value (Table 1). *Astronium urundeuva* (M.Allemão) Engl. has a higher redundancy value, while *Ceiba glaziovii* (Kuntze) K. Schum. has less redundancy (Table 1).

Phylogenetic clustering was strong for the species used for tool handles ($\lambda = 0.75$) and crafts ($\lambda = 1.01$) (Fig 2). All other usage categories showed weak phylogenetic clustering for firewood ($\lambda = 0.11$), household items, roof, hedge fencing, wire fencing and door ($\lambda = 6.67 \times 10^{-5}$), respectively.

We found a moderate degree of phylogenetic clustering for species redundancy ($\lambda = 0.78$, $p = 0.25$), but not significant, indicating that the redundant species were phylogenetically closer than expected. The phylogenetic clade containing *Schinopsis brasiliensis* and *Astronium urundeuva* presents the highest values for species redundancy (Fig 3A). An intermediate species redundancy was found in the clade containing the species of the *Mimosa* genus. On the other hand, *Amburana cearensis* and *Ceiba glaziovii*, which are exclusive species for each clade, have low redundancy (Fig 3A). The phylogenetic signal for the wood density of the cited species was weak ($\lambda = 0.24$, $p = 0.23$), indicating that the cited species with the wood density values are not phylogenetically close (Fig 3B). However, the clade containing *Tabebuia aurea* and *Handroanthus impetiginosus* are species that presented higher wood density.

The phylogenetic signal was weak when relating the wood density for each subcategory: firewood ($\lambda = 0.24$, $p = 0.26$), roof ($\lambda = 0.50$ $p = 0.73$), hedge fencing ($\lambda = 4.4382e-05$ $p = 1$), wire fencing ($\lambda = 0.07$, $p = 0.77$), door: ($\lambda = 4.26083e-05$, $p = 1$), household items ($\lambda = 1.69$, $p = 0.07$), crafts ($\lambda = 6.84751e-05$, $p = 1$) and tool handle ($\lambda = 0.01$, $p = 0.95$). Furthermore, we found no correlation between species redundancy and wood density ($\beta_{PGLS} = 0.45 \pm 0.47$, $t_{PGLS} = 0.94$, $p_{PGLS} = 0.34$).

## Discussion

The use of similar plants is a common activity among human populations [17, 46]. Although the causes that explain this phenomenon have been well evaluated in the literature [16, 47, 48], we are still moving towards understanding the influence of the evolutionary history of plants used by the traditional peoples of the Caatinga. Through our case study, we identified that the plant species cited for tool handles and handicrafts are phylogenetically close. Our results suggest that the characteristics of the species cited for these uses were similar. This becomes interesting because people over time may have realized that the characteristics of some species meet these demands. For example, it is expected that people look for resistant and strong plants for

**Table 1. Species cited for different usage categories by local populations in the state of Paraíba, Northeastern Brazil.** Followed by the vernacular name, Voucher represented by ACAM (Herbarium Manuel de Arruda Câmara), NC (not collected) and the values referring to the attribute (wood density) and index (species redundancy).

| Family | Species | vernacular name | Voucher | attribute | index |
|---|---|---|---|---|---|
| Anacardiaceae | *Astronium urundeuva* Allemão | Aroeira | ACAM 1991 | 0.55 | 2.89 |
| | *Schinopsis brasiliensis* Engl | Baraúna | ACAM 2009 | 0.47 | 2.67 |
| | *Spondias tuberosa* Arruda | Umbuzeiro | ACAM 1579 | 0.49 | 0.60 |
| Annonaceae | *Annona leptopetala* (R.E.Fr.) H. Rainer | Pinha-Brava | NC | 0.75 | 0.44 |
| Apocynaceae | *Aspidosperma pyrifolium* Mart. & Zucc. | Pereiro | ACAM 1995 | 0.61 | 2.51 |
| Bignoniaceae | *Handroanthus impetiginosus* (Mart. ex DC.) Mattos | Pau-d'arco-Rocho | NC | 0.83 | 1.09 |
| | *Tabebuia aurea* (Silva Manso) Benth. & Hook.f. ex S.Moore | Craibeira | NC | 1.03 | 2.04 |
| Boraginaceae | *Cordia trichotoma* (Vell.) Arráb. ex Steud. | Frei-Jorge | NC | 0.74 | 1.49 |
| Burseraceae | *Commiphora leptophloeos* (Mart.) J.B. Gillett | Umburana | NC | 0.43 | 1.71 |
| Capparaceae | *Cynophalla flexuosa* (L.) J. Presl | Feijão-Bravo | ACAM 2014 | 0.34 | 0.69 |
| Combretaceae | *Combretum leprosum* Mart. | Mufumbo | ACAM 1990 | 0.97 | 1.89 |
| Euphorbiaceae | *Cnidoscolus quercifolius* Pohl | Favela | ACAM 1996 | 0.44 | 0.20 |
| | *Croton blanchetianus* Baill. | Marmeleiro | NC | 0.67 | 2.29 |
| | *Croton heliotropiifolius* Kunth | Marmeleiro-branco | ACAM 1983 | 0.26 | 1.64 |
| | *Euphorbia tirucalli* L. | Aveloz | NC | 0.54 | 0.87 |
| | *Hymenaea courbaril* L. | Jatobá | NC | 0.95 | 0.42 |
| | *Jatropha mollissima* (Pohl) Baill. | Pinhão-Bravo | ACAM 1984 | 0.36 | 1.24 |
| | *Manihot glaziovii* Müll. Arg. | Maniçoba | NC | 0.35 | 0.38 |
| | *Sapium glandulosum* (L.) Morong | Burra-Leiteira | ACAM 2010 | 0.34 | 0.18 |
| Fabaceae | *Amburana cearensis* (Allemão) A.C.Sm. | Cumarú | ACAM 1981 | 0.6 | 0.11 |
| | *Anadenanthera colubrina* (Vell.) Brenan | Angico | ACAM 1982 | 0.88 | 1.87 |
| | *Bauhinia cheilantha* (Bong.) Steud | Mororó | ACAM 1986 | 0.91 | 2.27 |
| | *Libidibia ferrea* (Mart. ex Tul.) L.P.Queiroz var. *ferrea* | Jucá | ACAM 1996 | 0.97 | 2.22 |
| | *Cenostigma pyramidale* (Tul.) Gagnon & G.P. Lewis | Catingueira | ACAM 1988 | 0.11 | 2.09 |
| | *Chloroleucon foliolosum* (Benth.) G.P.Lewis | Jurema-de-coronha | ACAM 2011 | 0.83 | 1.71 |
| | *Dahlstedtia araripensis* (Benth.) M.J. Silva & A.M.G. Azevedo | Sucupira | ACAM 2007 | 0.62 | 0.73 |
| | *Enterolobium contortisiliquum* (Vell.) Morong | Tambor | ACAM 2006 | 0.52 | 0.80 |
| | *Erythrina velutina* Willd. | Mulungu | ACAM 2000 | 0.2 | 1.29 |
| | *luetzelburgia auriculata* (Allemão) Ducke | Pau-de-Serrote | ACAM 1997 | 0.36 | 0.62 |
| | *Mimosa arenosa* (Willd.) Poir. | Unha-de-gato | NC | 0.83 | 1.27 |
| | *Mimosa caesalpiniifolia* Benth | Sabiá | NC | 0.78 | 0.67 |
| | *Mimosa lewisii* Barneby | Jurema-amorosa | NC | 0.80 | 0.67 |
| | *Mimosa ophthalmocentra* Mart. ex Benth. | Jurema-de-embira | ACAM 1992 | 0.48 | 1.67 |
| | *Mimosa tenuiflora* (Willd.) Poir. | Jurema-Preta | ACAM 1989 | 0.60 | 2.07 |
| | *Piptadenia retusa* (Jacq.) P.G.Ribeiro, Seigler & Ebinger | Jurema-branca | ACAM 1978 | 0.61 | 1.67 |
| | *Senna spectabilis* (DC.) H.S.Irwin & Barneby | Canafístula | ACAM 2005 | 0.87 | 1.69 |
| Malvaceae | *Ceiba glaziovii* (Kuntze) K.Schum. | Barriguda | ACAM 1993 | 0.59 | 0.11 |
| Nyctaginaceae | *Guapira hirsuta* (Choisy) Lundell | João-Mole | NC | 0.48 | 0.67 |
| | *Guapira laxa* (Netto) Furlan | Pau-Piranha | NC | 0.48 | 0.60 |
| Olacaceae | *Ximenia americana* L. | Ameixa | NC | 0.83 | 1.38 |
| Rhamnaceae | *Sarcomphalus joazeiro* (Mart.) Hauenshild | Juazeiro | ACAM 1933 | 0.71 | 1.27 |
| Rubiaceae | *Coutarea hexandra* (Jacq.) K. Schum | Quina-Quina | NC | 0.6 | 1.02 |
| Rutaceae | *Zanthoxylum rhoifolium* Lam. | Limãzinho | NC | 0.25 | 0.67 |
| Sapotaceae | *Sideroxylon obtusifolium* (Roem. & Schult.) TDPenn. | Quixabeira | ACAM 1994 | 0.72 | 2.29 |

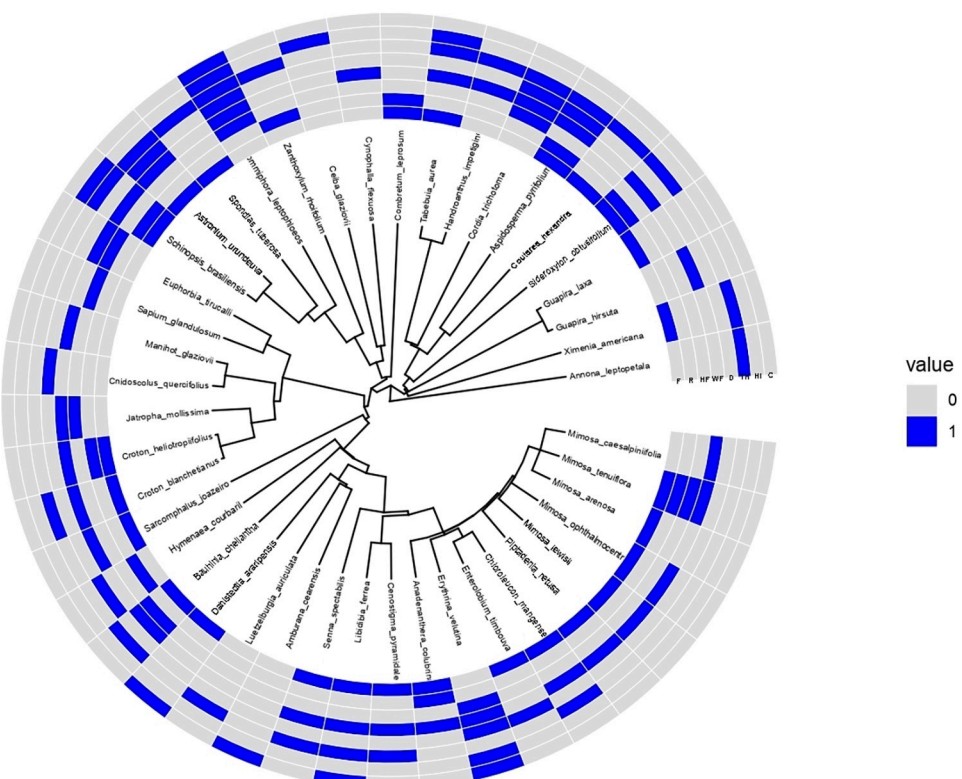

**Fig 2. Hypothetical phylogenetic tree of timber species cited by research participants.** This phylogenetic hypothesis was developed through the v. phylo.maker [Qian & Jin, 2019], and constructed from a comprehensive phylogeny with vascular plants from around the world [Qian & Jin, 2019]. The resulting ultrametric phylogenetic hypothesis has 44 tip labels and 43 internal nodes. The tree shows the evolutionary relationship of the mentioned plants for timber purposes. The uses are arranged in the blue color of the figure and correspond to the initials of each use, where: F (firewood), R (roof), HF (hedge fencing), WF (wire fencing), D (door), TH (tool handle), HI (household items) and C (crafts).

the use of tool handles considering that these artifacts are used for digging the soil, drilling walls and domestic cleaning.

Our results confirmed our prediction. The traits shared between the species mentioned for firewood and fences did not show a phylogenetic signal. Apparently this suggests that people tend to know less phylogenetically related species for these uses. Everything in the dry forest is useful in a situation of scarcity of species suitable for firewood. The literature shows that these uses demand greater amounts of plant biomass, as opposed to use for producing technological artifacts, which is a category that requires plants with lower biomass [8, 49]. This may explain the reasons why there was no phylogenetic clustering.

Alternatively, it is plausible to mention that given the abiotic filters imposed by seasonality and frequent droughts in dry forests [50], the flora presents a strong niche conservatism and life history convergence [51]. This convergence reflects the development of similar characteristics by distantly related species as an adaptive response to similar environmental pressures [52]. As a result of these similarities, perhaps even if people know less related plants in the present study, the similarities can be useful for the same activity.

The previous results are interesting from the point of view of plant selection, as they can present some information on people's behavior towards usage. For example, the social-ecological theory of maximization under the maximum environmental performance model predicts that people tend to incorporate useful plants which offer the maximum return among the

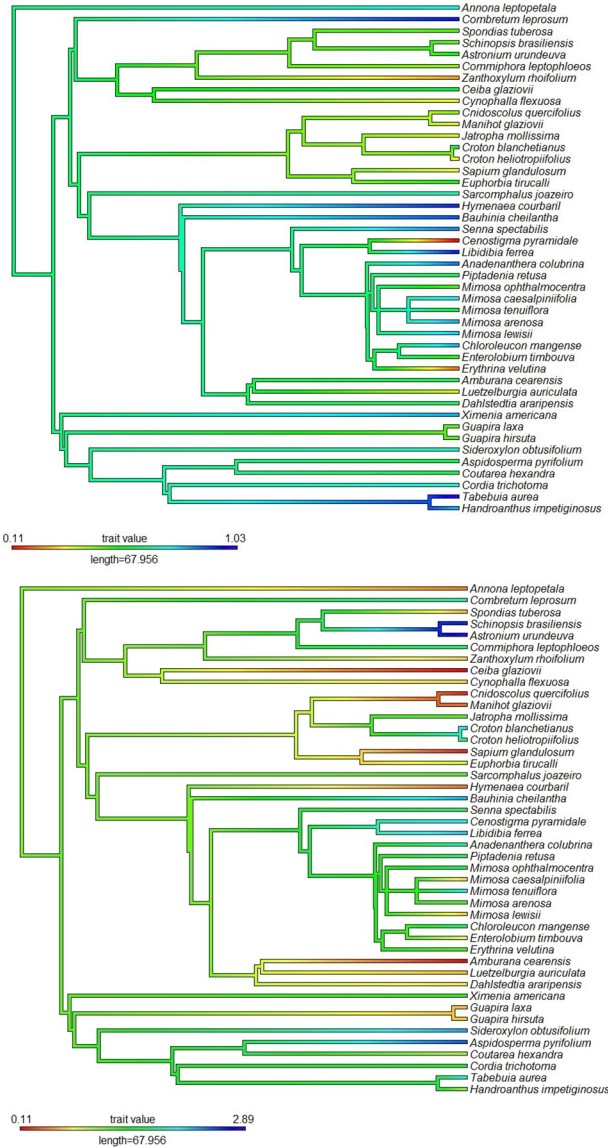

**Fig 3. Hypothetical phylogenetic representation of the species cited by local experts in a semi-arid region of the state of Paraíba, northeastern Brazil.** (A) Distribution of species redundancy values along the phylogeny. (B) Distribution of wood density values along the phylogenetic tree of 44 plants cited.

parameters that influence the use of resources [53]. Herein, we did not verify variables that may be influencing the collection of useful plant species, but we verified possible approximations that may be favoring the use of some species over others when controlling the evolutionary parentage of the mentioned plants. Thus, the strong phylogenetic signal in the characteristics of the species indicated for the uses of the present study may suggest that the local human populations have some specificity in handling the species. This specificity usually guides people towards expert behavior due to the non-random nature of selecting such resources [53]. On the other hand, the use of less related plants may suggest that people do not have specificity in choosing plants used as fences, firewood, roofs, doors and household items.

We also verified the importance of analyzing usage subcategories rather than general categories. Considering that each use of wood has different specificities (39), it is not interesting to verify the morphological traits and phylogenetic sign of plants cited for categories. We may lose information when considering the characteristics of the species cited in a general category. For example, people can mention a specific set of plants for the general category, but another set of species can be indicated for this when considering a subcategory. For example, [21] already observed that people tend to use fewer traits of palm species in the subcategories when compared to the general usage category.

Our findings also show that given the moderate phylogenetic signal in species redundancy, it is likely that the selection of redundant plants could be partly encouraged by evolutionary history. The cited species probably exhibit similar functional traits which in turn make them redundant. Furthermore, the relatedness of cited plants in the literature explains that therapeutically less redundant species do not always face greater usage pressure [18], but lower redundancy in our case study does not explain this. On the contrary, the few cited species included in the clades tend to indicate greater usage pressure on them. For example, we observed that *Amburana cearensis* has less redundancy, and this suggests that few species fulfill a certain function together with it.

Unlike studies that addressed utilitarian redundancy [21], herein we proposed to study the similarities in the functions of certain species cited by human groups [54]. Studying the redundancy of useful species is an important step towards understanding the selection and differential use of plants. For example, two species are functionally redundant in ecology if they have similar information in terms of values of a given attribute [55]; on the other hand, species are functionally similar in ethnobotany when they are cited in a given use.

The weak signal of the wood density of the species mentioned in our study draws attention. In an attempt to analyze whether the phylogenetic signal of wood density could be present among the use subcategories, we also found that it was not. This suggests that it was not possible to verify specificity between the wood density of the mentioned species and the usage subcategory in our study. This result reinforces that the plant selection criterion cannot only be limited to phylogeny, as we know that different variables can influence local use. For example, [12] observed positive relationships between wood density and people's perceived quality of species for biofuel use in the Caatinga. Thus, not including the perception of all research participants on the characteristics of the known species for each usage subcategory may have been a limitation in our study.

The lack of relationship between the wood density of the cited species and the subcategories may be further evidence of generality in the flora of the Caatinga, where most woody plants have high wood density [36], and this may have created noise in our analyses. The mentioned species which presented low wood density in our study, namely *Cenostigma pyramidale*, *Sapium glandulosum*, *Euphorbia tirucalli* and *Jatropha mollissima*, are those precisely cited by people for producing hedge fencing. Species with low wood density in the Caatinga are those with the greatest capacity to store water in their stems [36]. The characteristic of these species induces the ability to sprout and resistance to biodegradation [6]. This may explain the reason for the use of these species as hedge fencing through planting trees to demarcate small rural properties. On the other hand, species with higher wood density tend to be long-lived and have greater heights in dry forest regions [36]. *Astronium urundeuva*, *Tabebuia aurea*, *Libidibia ferrea* and *Bauhinia cheilantha* are some of the species with the highest wood density cited in our study.

## Conclusions

Wood from plants is one of the most used resources around the world. Human populations located in semi-arid environments collect wood from useful plants for domestic and rural activities and generally look for plants that are similar in morphological terms. This study highlighted the importance of controlling the evolutionary history of useful plants cited by local populations inserted in a Caatinga region. The phylogenetic component was an important resource for discovering plants non-randomly used for tool handles and handicrafts, considering that the mentioned plants present phylogenetic proximity. In contrast, the plants cited for firewood and fences do not present a phylogenetic signal. Moreover, we found a moderate degree of phylogenetic clustering for species redundancy and weak for wood density. The phylogenetic proximity of useful plants and the lower redundancy for some species in our study may suggest greater use pressure, given that few species fulfill the same function. Our results fill a gap on how the evolutionary relationship of useful plants can serve to understand plant selection by human groups in the Caatinga. However, we recommend that future studies aim to understand whether there are characteristics grouped within phylogenies that can guide human behavior in plant selection.

## Supporting information

**S1 Table. List with the name of the species and wood usage considered in the study, and the works consulted to collect data on wood density.**
(DOCX)

## Acknowledgments

The authors would like to thank the local populations of the five rural communities for kindly sharing valuable information on the use of plants in the Caatinga. Special thanks to Sonally S. Cunha, Maria Graciele M. Rodrigues, Stefanny M. Souza, Wendell F. S. Gaudêncio and all the members of the Laboratory of Neotropical Ecology at the State University of Paraíba for their support in the fieldwork. To Professor Dr. José Iranildo Miranda de Melo for his assistance in identifying botanical species. To Professor Dr. Danilo M. Neves for reading and suggestions in writing the article. Hyago K. Lucena for providing some of the codes used in phylogenetic analysis.

## Author Contributions

**Conceptualization:** Kamila Marques Pedrosa.

**Data curation:** Kamila Marques Pedrosa.

**Formal analysis:** Kamila Marques Pedrosa.

**Funding acquisition:** Sérgio de Faria Lopes.

**Investigation:** Kamila Marques Pedrosa, Maiara Bezerra Ramos.

**Methodology:** Kamila Marques Pedrosa, Maiara Bezerra Ramos.

**Project administration:** Kamila Marques Pedrosa, Maiara Bezerra Ramos.

**Supervision:** María de los Ángeles La Torre-Cuadros, Sérgio de Faria Lopes.

**Writing – original draft:** Kamila Marques Pedrosa.

**Writing – review & editing:** Maiara Bezerra Ramos, María de los Ángeles La Torre-Cuadros, Sérgio de Faria Lopes.

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
