## [Decision Letter · Decision Letter 0]

27 Feb 2023

PONE-D-23-02443Plant parentage influences the type of timber use by traditional peoples of the Brazilian CaatingaPLOS ONE

Dear Dr. Lopes,

Thank you for submitting your manuscript to PLOS ONE. After careful consideration, we feel that it has merit but does not fully meet PLOS ONE’s publication criteria as it currently stands. Therefore, we invite you to submit a revised version of the manuscript that addresses the points raised during the review process.

We look forward to receiving your revised manuscript.

Kind regards,

Tzen-Yuh Chiang

Academic Editor

PLOS ONE

“The authors would like to thank the local populations of the five rural communities for kindly sharing valuable information on the use of plants in the Caatinga. Special thanks to Sonally S. Cunha, Maria Graciele M. Rodrigues, Stefanny M. Souza, Wendell F. S. Gaudêncio and all the members of the Laboratory of Neotropical Ecology at the State University of Paraíba for their support in the fieldwork. To Professor Dr. José Iranildo Miranda de Melo for his assistance in identifying botanical species. To Professor Dr. Danilo M. Neves for reading and suggestions in writing the article. Hyago K. Lucena for providing some of the codes used in phylogenetic analysis. MBR and KMP thank the Fundação de Apoio à Pesquisa do Estado da Paraíba (FAPESq) for granting research grants. This study was partly funded by the State University of Paraíba grant 03/2022. SFL thanks CNPq for the productivity grant.”

“MBR and KMP acknowledge the Fundação de Apoio à Pesquisa do Estado da Paraíba (FAPESq) for granting research scholarships.  SFL thanks CNPq for a productivity grant.

https://fapesq.rpp.br/

NO - Include this sentence at the end of your statement: The funders had no role in study design, data collection and analysis, decision to publish, or preparation of the manuscript.”

Reviewers' comments:

Reviewer's Responses to Questions

**Comments to the Author**

1. Is the manuscript technically sound, and do the data support the conclusions?

Reviewer #1: Yes

2. Has the statistical analysis been performed appropriately and rigorously? 

Reviewer #1: Yes

3. Have the authors made all data underlying the findings in their manuscript fully available?

Reviewer #1: Yes

4. Is the manuscript presented in an intelligible fashion and written in standard English?

Reviewer #1: Yes

5. Review Comments to the Author

Reviewer #1: The recent disturbance of ecosystem services by anthropogenic sources has increases the needs to use local plant for social and cultural purpose. The uses of such species in many cases are difficult to understand and are most used interchangeably. The current manuscript entitled “Plant parentage influences the type of timber use by traditional peoples of the Brazilian Caatinga” is a quality research conducted by the authors highlighting one such issue. I appreciated the hard work and research concerns of the authors. However, despite the importance of the idea, the manuscript language in most section is confused and seems very poor, although I am not native to English. Moreover, I highlighted some changes in different sections of the manuscript, which may further aid in the quality of the manuscript.

Abstract

The abstract needs some revisions for a clear understanding of the readers. Some specific comments are

- Line 20-21. Please remove this sentence as it seems out of context

-Line 25-27. Revise the lines please very vogue

-Line 27-28. Is it necessary to have hypothesis in abstract? Better to avoid such things in abstract

-Line 29. More like conclusion, if it’s added at the end of abstract as conclusion

-Conclusion missing

Introduction

The introduction is to the point but need language improvement in the sentence structure and also looks out for topographic mistakes. Some specific comments are

-Line 54-55. Remove unnecessary words i.e. “in the first case”

-Line 64.. by chance???is it necessary or important to mention

-Line 82-83. Please revise very vogue

Material and method

Well and comprehensive written but need topographic correction. In addition, it lacks informations on the soil texture, electrical conductivity, and nutrient like organic matter and nitrogen which play critical role in the availability and accumulation of the heavy metals.

-Line 95. Revise please, very unnecessary verbs are used

-Line 103. Past tense please, its manuscript methodology

-Line 130-131. Please revise the sentence

-Line 134-135. Very unnecessary sentence

-Line 136-143. Please properly introduce your questionnaire and interview section i.e. classify the question section wise.. Like respondents demography, Social status, question about research etc.

-Line 166-167. Revise the sentence please

Results

-The portion is satisfactory, check for minor topographic corrections

Discussion

The Discussion is to the point, but need language improvement for better understanding of the readers. Some specific comments are

Page 16. As in our study, (44) observed that useful woody plants, phylogenetically close, located on the Amalfi Coast in Italy, were preferred for domestic practices due to tool production. (Please revise)

Page 17. Our results confirmed our prediction. The traits shared between the species mentioned for firewood and fences did not show a phylogenetic signal. Apparently this suggests that people tend to know less phylogenetically related species for these uses. Everything in the dry forest is useful in a situation of scarcity of species suitable for firewood. The literature shows that these uses demand larger amounts of plant biomass than categories that require less biomass, such as plants used in the production of technological artifacts (8,45). This may explain the reasons why there was no phylogenetic clustering.

(Please revise, very poor writing)

Page 17. Same issue with the next paragraph

Page 18. Unlike studies that addressed utilitarian redundancy (Albuquerque & Oliveira, 2006),

Correct the citation please

Conclusion

The conclusion needs to be refined and provide recommendations for further studies. In addition try to reduce the bulk of result repetition in conclusion.

6. PLOS authors have the option to publish the peer review history of their article (what does this mean?). If published, this will include your full peer review and any attached files.

Reviewer #1: No

---

## [Author Response · Author response to Decision Letter 0]

13 Apr 2023

To: Dr. Tzen-Yuh Chiang

Academic Editor of PLOS ONE

Thank you for potentially accepting our manuscript entitled “Plant parentage influences the type of timber use by traditional peoples of the Brazilian Caatinga” for publication in PLOS ONE.

We would like to acknowledge the excellent suggestions from both reviewers for improving the revised version of our manuscript.

The suggested edits and comments made by the reviewers are shown below.

Reviewers' comments:

Response: done.

Response: Thank you for the instructions. We certify that the paper is in the style of PLOS ONE. We removed the funding from the acknowledgements session. Below we describe our funding source:

Funding Statement: MBR and KMP thank the Fundação de Apoio à Pesquisa do Estado da Paraíba (FAPESq) for granting research grants. This study was partly funded by the State University of Paraíba grant 03/2022. SFL thanks CNPq for the productivity grant.

Response: Thank you for the instructions. The data generated and analyzed for the current study has been uploaded to the Dryad Digital Repository https://doi.org/10.5061/dryad.w3r2280w5

Response: Thank you for the instructions. The satellite images were obtained from the database of AESA (Agência Nacional das Águas da Paraíba, Brazil). AESA uses the satellite images from Google Earth. We contacted the agency and explained about the Creative Commons Attribution License (CC BY 4.0), but it was explained to us that the satellite images used are not copyrighted and are in the public domain. The figure inserted next to the map is by the authors.

Response: Thank you for the instructions. The information was inserted in the text.

Reviewer #1 done 

Abstract

The abstract needs some revisions for a clear understanding of the readers. Some specific comments are:

1) Line 20-21. Please remove this sentence as it seems out of context

Response: done

2) Line 25-27. Revise the lines please very vogue

Response: done

3) Line 27-28. Is it necessary to have hypothesis in abstract? Better to avoid such things in abstract

Response: Thanks for the suggestion. The hypothesis mentioned in the text of the abstract is related to the method. To build our phylogenetic tree we use a published tree. In evolution is called phylogenetic hypothesis.

4) Line 29. More like conclusion, if it’s added at the end of abstract as conclusion

Response: done

5) Conclusion missing 

Response: Thank you for the suggestion. The text of the conclusion has been rewritten.

Introduction

6) The introduction is to the point but need language improvement in the sentence structure and also looks out for topographic mistakes. Some specific comments are

7) Line 54-55. Remove unnecessary words i.e. “in the first case”

Response: Done.

8) Line 64. By chance?? Is it necessary or important to mention.

Response: Yes. “by chance” makes reference to evolving Brownian motion. 

9) Line 82-83. Please revise very vogue

Response: Thanks for the suggestion. The hypothesis mentioned in the text of the abstract is related to the method. To build our phylogenetic tree we use a published tree. In evolution is called phylogenetic hypothesis.

Material and method

10) Well and comprehensive written but need topographic correction. In addition, it lacks informations on the soil texture, electrical conductivity, and nutrient like organic matter and nitrogen which play critical role in the availability and accumulation of the heavy metals.

Response: Thanks for the suggestion. the information has been added.

11) Line 95. Revise please, very unnecessary verbs are used.

Response: Done.

12) Line 103. Past tense please, its manuscript methodology.

Response: Done.

13) Line 130-131. Please revise the sentence.

Response: Done.

14) Line 134-135. Very unnecessary sentence.

Response: Thanks for the suggestion. This information is important to detail the local ecological knowledge data.

15) Line 136-143. Please properly introduce your questionnaire and interview section i.e. classify the question section wise. Like respondents demography, social status, question about research etc.

Response: Thanks for the suggestion. A new subtopic has been inserted to present the interviews.

16) Line 166-167. Revise the sentence please

Response: Done.

Results

17) The portion is satisfactory, check for minor topographic corrections

Response: Done.

Discussion

The discussion is to the point, but need language improvement for better understanding of the readers. Some specific comments are.

Response: Thank you for your suggestion. The text has been revised to improve the language and a better understanding for the readers.

18) Page 16. As in our study, (44) observed that useful woody plants, phylogenetically close, located on the Amalfi Coast in Italy, were preferred for domestic practices due to tools production. (Please revise)

Response: Thanks for the suggestion. The quote would be to compare the results. Since it did not allow for contextualization we decided to remove it.

19) Page 17. Our results confirmed our prediction. The traits shared between the species mentioned for firewood and fences did not show a phylogenetic signal. Apparently this suggests that people tend to know less phylogenetically related species for these uses. Everything in the dry forest is useful in a situation of scarcity of species suitable for firewood. The literature shows that these uses demand larger amounts of plant biomass than categories that require less biomass, such as plants used in the production of technological artifacts (8,45). This may explain the reasons why there was no phylogenetic clustering. (Please revise, very poor writing).

Response: Thanks for the suggestion. Done.

20) Page 17. Same issue with the next paragraph.

Response: Done.

21) Page 18. Unlike studies that addressed utilitarian redundancy (Albuquerque & Oliveira 2006).

Correct the citation please. 

Response: Done.

Conclusion

The conclusion needs to be refined and provide recommendations for further studies. In addition try to reduce the bulk of result repetition in conclusion.

Response: Thanks for the suggestion. Done.

Note: In addition to the reviewers' suggestions, we have inserted some excerpts, which are highlighted, in the text. In addition, we have changed the position of the topic.

Page 3 (Line 54-56)

Page 4 (Line 84)

Page 6 (Line 129-140)

Page 9 (Line 207)

Page 12 (Table 1)

Page 22 (discussion)

Page 23 (conclusions)

Page 25- 32 (references)

Kamila Marques Pedrosa

State University of Paraíba, Department of Biology, Campina Grande, Paraíba, Brazil.

---

## [Decision Letter · Decision Letter 1]

18 Apr 2023

PONE-D-23-02443R1Plant parentage influences the type of timber use by traditional peoples of the Brazilian CaatingaPLOS ONE

Dear Dr. Lopes,

Thank you for submitting your manuscript to PLOS ONE. After careful consideration, we feel that it has merit but does not fully meet PLOS ONE’s publication criteria as it currently stands. Therefore, we invite you to submit a revised version of the manuscript that addresses the points raised during the review process.

We look forward to receiving your revised manuscript.

Kind regards,

Tzen-Yuh Chiang

Academic Editor

PLOS ONE

Journal Requirements:

Reviewers' comments:

Reviewer's Responses to Questions

**Comments to the Author**

1. If the authors have adequately addressed your comments raised in a previous round of review and you feel that this manuscript is now acceptable for publication, you may indicate that here to bypass the “Comments to the Author” section, enter your conflict of interest statement in the “Confidential to Editor” section, and submit your "Accept" recommendation.

Reviewer #1: (No Response)

2. Is the manuscript technically sound, and do the data support the conclusions?

Reviewer #1: Yes

3. Has the statistical analysis been performed appropriately and rigorously? 

Reviewer #1: Yes

4. Have the authors made all data underlying the findings in their manuscript fully available?

Reviewer #1: Yes

5. Is the manuscript presented in an intelligible fashion and written in standard English?

Reviewer #1: Yes

6. Review Comments to the Author

Reviewer #1: Abstract

Comments 1. Line 25-27 previously and now 23-25: We estimated the redundancy values for the mentioned species, and we compiled data from the literature on the wood density values of the cited species through a literature search.

Revise like… We estimated the redundancy values of the species and compiled data from the literature on the wood density values of the cited species.

Line 30-32: Not concluded properly

In addition, the language need thoroughly editing for improvement of the text

7. PLOS authors have the option to publish the peer review history of their article (what does this mean?). If published, this will include your full peer review and any attached files.

Reviewer #1: **Yes: **Rafi Ullah

---

## [Author Response · Author response to Decision Letter 1]

10 May 2023

The suggested edits and comments made by the reviewers are shown below.

Reviewer #2 done 

Abstract

Comments 1. Line 25-27 previously and now 23-25: We estimated the redundancy values for the mentioned species, and we compiled data from the literature on the wood density values of the cited species through a literature search.

Revise like… We estimated the redundancy values of the species and compiled data from the literature on the wood density values of the cited species.

Response: Thank you for the suggestion. Done.

Line 30-32: Not concluded properly

In addition, the language need thoroughly editing for improvement of the text

Response: Thank you for the suggestion. Done.

---

## [Decision Letter · Decision Letter 2]

16 May 2023

Plant parentage influences the type of timber use by traditional peoples of the Brazilian Caatinga

PONE-D-23-02443R2

Dear Dr. de Faria Lopes,

We’re pleased to inform you that your manuscript has been judged scientifically suitable for publication and will be formally accepted for publication once it meets all outstanding technical requirements.

Kind regards,

Tzen-Yuh Chiang

Academic Editor

PLOS ONE

Additional Editor Comments (optional):

Reviewers' comments:

Reviewer's Responses to Questions

**Comments to the Author**

1. If the authors have adequately addressed your comments raised in a previous round of review and you feel that this manuscript is now acceptable for publication, you may indicate that here to bypass the “Comments to the Author” section, enter your conflict of interest statement in the “Confidential to Editor” section, and submit your "Accept" recommendation.

Reviewer #1: All comments have been addressed

2. Is the manuscript technically sound, and do the data support the conclusions?

Reviewer #1: Yes

3. Has the statistical analysis been performed appropriately and rigorously? 

Reviewer #1: Yes

4. Have the authors made all data underlying the findings in their manuscript fully available?

Reviewer #1: Yes

5. Is the manuscript presented in an intelligible fashion and written in standard English?

Reviewer #1: Yes

6. Review Comments to the Author

Reviewer #1: The authors has substantially revises the manuscript (Plant parentage influences the type of timber use by traditional peoples of the Brazilian Caatinga") according to the suggestion and can be accept for publication.

Thank you

Dr. Rafi Ullah

7. PLOS authors have the option to publish the peer review history of their article (what does this mean?). If published, this will include your full peer review and any attached files.

Reviewer #1: **Yes: **Rafi Ullah

---

## [Editor Report · Acceptance letter]

8 Jun 2023

PONE-D-23-02443R2 

Plant parentage influences the type of timber use by traditional peoples of the Brazilian Caatinga 

Dear Dr. Lopes:

I'm pleased to inform you that your manuscript has been deemed suitable for publication in PLOS ONE. Congratulations! Your manuscript is now with our production department. 

Kind regards, 

on behalf of

Dr. Tzen-Yuh Chiang 

Academic Editor

PLOS ONE